# Anti-*Helicobacter pylori* Compounds of *Sambucus williamsii* Hance Branch

**DOI:** 10.3390/plants14162558

**Published:** 2025-08-17

**Authors:** Woo-Jin Jeong, Dong-Min Kang, Atif Ali Khan Khalil, Bashu Dev Neupane, Seong-Joon Cho, Na-In Yang, Ki-Hyun Kim, Mi-Jeong Ahn

**Affiliations:** 1College of Pharmacy and Research Institute of Pharmaceutical Sciences, Gyeongsang National University, Jinju 52828, Republic of Korea; 8.27mm@gnu.ac.kr (W.-J.J.); kdm7105@gnu.ac.kr (D.-M.K.); neupanebashudev@gnu.ac.kr (B.D.N.); 2024210438@gnu.ac.kr (S.-J.C.); 2Department of Biotechnology, Yeungnam University, Gyeongsan 38541, Republic of Korea; atif.khalil7799@gmail.com; 3School of Pharmacy, Sungkyunkwan University, Suwon 16419, Republic of Korea; amellia00@skku.edu

**Keywords:** *Sambucus williamsii*, anti-*Helicobacter pylori* activity, guaiacylglycerol, boehmenan, lignans

## Abstract

*Sambucus williamsii* Hance (Viburnaceae), the Korean elderberry, is widely used in herbal medicine and in the food industry. It is known to have various pharmacological effects, including antitumor, antioxidant, anti-inflammatory, and antimicrobial activities. During our search for anti-*Helicobacter pylori* compounds from natural resources, the methanol extract of the *S. williamsii* branch significantly inhibited the growth of *H. pylori*. Three phenolic and four lignan compounds were isolated from the methylene chloride fraction that had shown the most potent anti-*H. pylori* activity among the hexane, methylene chloride, ethyl acetate, butanol, and water fractions. The chemical structures were identified to be three phenolics of sylvopinol (**1**), dihydroconiferyl alcohol (**2**), and (7*S*,8*R*)-guaiacylglycerol (**3**) and four lignans of boehmenan (**4**), (7*S*,8*S*)-guaiacylglycerol *β*-coniferyl ether (**6**) and lawsonicin (**7**) with a new lignan, (7*R*,8*R*)-sambucanol (**5**), the structure of which was established by ^1^H- and ^13^C-NMR, and HRESI-MS, as well as quantum chemical electronic circular dichroism (ECD) calculations. Among the isolates, compounds **3** and **4** exhibited significant anti-*H. pylori* activity against strains 51 and 26695. Compound **3** displayed more potent antibacterial activity with MIC values of 3.13 and 6.25 μM, and MIC_50_ values of 28.5 and 56.8 μM against the two strains, respectively. Their inhibitory activities were higher than those of a positive control, quercetin. Furthermore, these two compounds showed moderate urease inhibitory activity. A molecular docking simulation revealed the high binding ability of **3** and **4** to the active site of *H. pylori* urease. These results will provide further insights into the design of more potent natural products for eradicating *H. pylori*.

## 1. Introduction

Gastric cancer ranks as the fifth most prevalent disease globally, characterized by a correspondingly high mortality rate. Notably, as of 2020, East Asian countries exhibit the highest incidence of gastric cancer worldwide [1,2]. The primary causes of gastric cancer include dietary and lifestyle factors, stress, and various other influences. Meanwhile, infection with *Helicobacter pylori* is considered to be the most significant and direct causative factor [3]. *H. pylori* is a spiral-shaped, Gram-negative bacterium that colonizes the stomachs of humans and animals. It was discovered in 1983 by J. Robin Warren and Barry J. Marshall [4]. Although the stomach is expected to be inhospitable to microorganisms due to its firm acidity, *H. pylori* possesses urease, which allows it to neutralize gastric acid and inhabit the mucosal layer and the mucus [5]. More than half of the global population is infected with *H. pylori*, with over 50% of the adult population in the Republic of Korea reported to be infected [6]. This bacterium is responsible for various gastrointestinal disorders, including gastric cancer, gastric ulcers, gastritis, and gastric MALT lymphoma. The International Agency for Research on Cancer has classified *H. pylori* as a Group 1 carcinogen [7,8]. Upon invading the gastric environment, *H. pylori* produces several inflammatory substances, including urease, CagA, exotoxin (VacA), and lipopolysaccharide (LPS) [9]. Urease, a ureolytic enzyme, hydrolyzes urea into carbon dioxide and ammonia. By generating urease while residing between the gastric mucosal and mucus layers, *H. pylori* alters the pH of the gastric mucosa, undermining its defensive mechanisms [10,11].

Current treatment options for *H. pylori* infection include triple therapy, which combines proton pump inhibitors (PPIs) with amoxicillin, clarithromycin, or metronidazole, and quadruple therapy, which consists of PPIs, bismuth, metronidazole, and tetracycline. However, there is a growing global concern regarding antibiotic resistance and associated side effects, leading to a decrease in treatment efficacy [12,13,14]. Consequently, the need for new drug development is increasing, and there is a rising interest in natural products-based pharmaceuticals, which are associated with relatively fewer side effects than synthetic drugs.

*Sambucus williamsii* Hance, commonly known as Korean elderberry, belongs to the family Viburnaceae. This deciduous shrub is widely distributed in East Asia, including Korea, China, and Japan, typically reaching heights of 5 to 6 m. The leaves are pinnately compound, featuring a palmate arrangement. The flowers are arranged in panicles and bloom in ivory-colored clusters between April and May. The fruits, which are either red or black, mature in July [15]. Traditionally, the stems and branches of this plant and its related species have been utilized in folk medicine under the name “Jeop-Gol-Mok”, primarily for the treatment of fractures [16]. In a previous study, the CHCl_3_ fraction of *S. ebulus* leaves and the fruit extract of American elderberry showed moderate anti-*H. pylori* activity [17,18]. Moreover, *S. williamsii* and *S. pendula* demonstrated more potent antibacterial activity against Gram-negative bacteria (*Escherichia coli*, *Pseudomonas aeruginosa*, *Pseudomonas fluorescens,* and *Salmonella typhimurium*) than Gram-positive bacteria (*Bacillus cereus*, *Bacillus subtilis,* and *Listeria monocytogenes*) [19]. Despite their antibacterial activity, there is no research on the anti-*H. pylori* compounds of *Sambucus* species or anti-*H. pylori* activity of *Sambucus* species branches.

To date, various classes of compounds have been isolated from the stems and branches of *S. williamsii*, including lignans, terpenoids, iridoids, and phenolics [20]. Studies have demonstrated significant biological activities associated with these compounds, encompassing anti-inflammatory, antibacterial, anticancer, anti-aging, antioxidant, and gastroprotective effects [20,21,22,23,24]. The present study aims to evaluate anti-*H. pylori* activity of *S. williamsii* and identify the active compounds as potential agents for treatment of *H. pylori*.

## 2. Results and Discussion

The branches of *S. williamsii* were extracted with 100% methanol, and the extract was fractionated according to the solvent polarity (hexane, CH_2_Cl_2_, EtOAc, *n*-BuOH, and water). Among the fractions, the CH_2_Cl_2_ fraction showed the most potent anti-*H. pylori* activity against the two strains 51 and 26695 (Table 1). To identify the anti-*H. pylori* substituents of *S. williamsii*, seven compounds (**1**−**7**) were isolated from the CH_2_Cl_2_ fraction by open column chromatography (CC) using Sephadex LH-20 resin, medium-pressure liquid chromatography (MPLC) and preparative high-performance liquid chromatography (HPLC) with a reversed-phase C18 column (Figure 1 and Appendix A). The chemical structures of these compounds were determined based on ESI-QTOF-MS, ^1^H- and ^13^C-NMR, and ECD spectroscopy data (Table 2, Figure 2 and Appendix A).

### 2.1. Structural Elucidations of the Isolated Compounds ***1***–***7***

Compound **5** was obtained as a white amorphous powder. In the ESI-QTOF-MS spectrum, a quasimolecular ion peak at *m*/*z* 399.1379 corresponding to [M + Na]^+^ was observed, suggesting a molecular formula of C_20_H_24_O_7_ (calcd. C_20_H_24_O_7_Na, 399.1412). The ^1^H-NMR spectrum revealed proton signals for a 1,3,5- and a 1,2,4-trisubstituted aromatic rings at δ 7.01 (2H, d, *J* = 1.4 Hz, H-4, 6) and 6.88 (2H, d, *J* = 1.4 Hz, H-2, 2′), 6.83 (1H, dd, *J* = 8.1, 1.4 Hz, H-6′) and 6.73 (1H, d, *J* = 8.1 Hz, H-5′) ppm, indicating meta- and ortho-coupling, respectively. The signals at *δ* 6.53 (1H, dt, *J* = 15.9, 1.4 Hz, H-7′) and 6.25 (1H, td, *J* = 15.9, 5.7 Hz, H-8′) ppm were attributed to two protons coupled to a trans double-bond. Additionally, the ^1^H-NMR spectrum showed peaks at *δ* 4.84 (1H, d, *J* = 5.7 Hz, H-7) and 4.37 (1H, td, *J* = 5.7, 3.8 Hz, H-8) ppm, corresponding to oxymethine protons. The peaks at *δ* 3.82 (3H, s, 3-OCH_3_) and 3.81 (3H, s, 3′-OCH_3_) ppm were assigned to the two methoxy groups. The ^13^C-NMR spectrum revealed a total of 20 carbon peaks, including 12 corresponding to the two aromatic rings. The peaks at *δ* 72.8, 62.4, and 60.8 ppm indicated the presence of three hydroxyl-substituted carbons, while those at *δ* 55.1 and 55.9 ppm were attributed to the two methoxy groups. These NMR data of compound **5** exhibited a similar pattern to those of an 8-*O*-4′-neolignan glucoside compound **4a** (**4b**, a peracetylated derivative) isolated from *Arum italicum* [25]. Structurally, the reference compound **4a** possesses an *O*-*β*-D-glucose moiety at the C-5 position and a 7*S*,8*S*-configuration based on NMR and CD analysis (the positive Cotton effect). In contrast, compound **5** corresponds to the aglycone form, lacking the glucose moiety. In this study, the absolute configuration of compound **5** was determined using ECD spectroscopy (Figure 3). The negative CD curve [λ_max_ 251 nm (Δε −0.06), λ_max_ 279 nm (Δε −0.07)] observed in the ECD spectrum of compound **5** indicated that compound **5** has a 7*R*,8*R*-configuration, which is distinct from the 7*S*,8*S*-configuration of compound **4a [25]**. To verify the absolute configuration of compound **5**, two possible isomers, (7*R*,8*R*)-compound **5** and (7*S*,8*S*)-compound **5**, were used for ECD calculations, and the experimental ECD spectrum of compound **5** was in good agreement with the calculated ECD curve of (7*R*,8*R*)-compound **5** (Figure 3), indicating the absolute configuration of C-7 and C-8 could be verified as *R* and *R*, respectively. Based on these spectroscopic results and comparison with reported data, compound **5** was designated as (7*R*,8*R*)-sambucanol, a new neolignan compound that was identified for the first time in this study. Although compound 6 had been previously reported as guaiacylglycerol *β*-coniferyl ether, its stereochemical configuration had not been determined prior to this study [26].

The ECD spectrum of compound **6** showed a positive Cotton effect at 217 nm [λ_max_ 217 nm (Δε +0.06)], supporting the assignment of its absolute stereochemistry as a 7*S*,8*S*-configuration [27]. The absolute configuration of compound **6** was confirmed using ECD calculations. The experimental ECD spectrum of compound **6** showed almost the same tendency as that of (7*S*,8*S*)-guaiacylglycerol *β*-coniferyl ether (Figure 3), which suggested that the absolute configuration of compound **6** was 7*S*,8*S*. The other five compounds were also isolated from this fraction and identified as the three phenolic compounds of sylvopinol (**1**) [28], dihydroconiferyl alcohol (**2**) [29], and (7*S*,8*R*)-guaiacylglycerol (**3**) [30], and the other two lignans of boehmenan (**4**) [31] and lawsonicin (**7**) [32] by comparison of their spectroscopic data, such as MS and NMR, with the published literature.

### 2.2. Anti-H. pylori Activity of Compounds ***1***−***7***

The seven isolated compounds were evaluated for their anti-*Helicobacter pylori* activity (Table 3). Among them, compounds **1**, **3**, and **4** showed noticeable antibacterial effects, with compounds **3** and **4** exhibiting particularly strong activity. To further investigate their potential, the minimal inhibitory concentration (MIC) values of compounds **3** and **4** were determined against two strains, 51 and 26,695, using the broth dilution method based on a previously reported protocol [33]. As shown in Table 4, both compounds showed significant growth inhibition in a concentration-dependent manner. For compound **3**, the MIC_50_ values were 28.5 μM and 56.8 μM against strains 51 and 26,695, respectively, while the MIC_90_ values were 97.1 μM and 86.5 μM. Compound **4** also showed inhibitory effects with the MIC_50_ values of 66.0 μM and 62.0 μM against the two strains.

Although the two phenyl propane compounds **2** and **3** have the same skeleton, they exhibited quite significant differences in their bacterial growth inhibitory activity. Compound **3**, which contains each hydroxyl group at C-7 and C-8, exhibited much higher inhibitory activity than compound **2**, which has no substitution at C-7 and C-8. This result suggests that the two hydroxyl groups at the C-7 and C-8 positions of **3** increase the inhibitory activity of *H. pylori*. It is known that the presence of polar functional groups such as hydroxyl and carboxyl moieties increased antibacterial activity [34].

Among the four neolignan isolates (**4**−**7**), only boehmeman (**4**) exhibited significantly potent anti-*H. pylori* activity. This compound contains two ferulic acid moieties and one oxolane moiety, which may contribute to its enhanced activity. Previous studies have reported the moderate anti-*H. pylori* activity of ferulic acid [35,36], and it is noteworthy that this moiety is present only in compound **4** among the four lignans. In another study, researchers modified the structure of lignans by introducing acetyl, oxolane, and hydroxymethyl groups, and evaluated their antibacterial activity against *Bacillus aryabhattai* and *Klebsiella* species. Among these modifications, lignans containing the oxolane moiety exhibited the most potent antibacterial growth inhibitory activity [37]. Generally, the antimicrobial properties of lignans are attributed to their ability to penetrate bacterial cell walls, resulting in cell wall lysis. This disruption induces oxidative stress, ATP depletion, and a decrease in intracellular pH within bacterial cells, ultimately compromising cellular integrity and leading to cell death [38].

Although compound **6** shares the guaiacylglycerol moiety with compound **3**, which showed potent anti-*H. pylori* activity, its inhibitory activity was very low. This discrepancy can be ascribed to the differing stereochemistry—compound **6** has a stereochemistry of (7*S*,8*S*), while compound **3** has (7*S*,8*R*). Additionally, compound **6** contains a coniferyl alcohol moiety at the C-7 position. Unlike ferulic acid, coniferyl alcohol, one of the major precursors of lignans, exhibited very weak antibacterial activity, and there are no reports on its anti-*H. pylori* activity [39]. Similarly, compound **5**, which also possesses a coniferyl alcohol moiety, failed to inhibit the bacterial growth.

Our findings provide a valuable basis for further in vitro and in vivo validation studies, which may ultimately advance these isolated compounds closer to preclinical evaluation and clinical translation as natural product-derived therapeutics, increasing the challenges created by *H. pylori* developing resistance to conventional antibiotics.

### 2.3. Urease Inhibitory Activity

Urease is recognized as a key virulence factor of *H. pylori*, catalyzing the hydrolysis of urea into ammonia and carbon dioxide, thereby neutralizing gastric acid and facilitating bacterial survival in the acidic environment of the stomach [10,11]. Therefore, the urease inhibitory activity of compounds **3** and **4**, which showed high growth-inhibitory efficacy against *H. pylori* strains 51 and 26,695, was evaluated. The results revealed that both compounds exhibited moderate inhibition activity (Table 5). The urease inhibitory activity (38.9 ± 1.2%) of compound **4** at the final concentration of 1 mM was higher than that (31.8 ± 2.8%) of compound **3**. At this condition, the inhibitory activity of acetohydroxamic acid, a positive control, was 73.5 ± 3%. The anti-urease activity of these two compounds is reported for the first time. Anti-*H. pylori* urease activity of ferulic acid has been reported, and compound **4** possesses two ferulic acids [40]. So, the moderate anti-urease activity of compound **4** could be derived from the two ferulic acid moieties.

### 2.4. Molecular Docking Simulation

Molecular docking is a valuable technique that can accurately predict the optimal orientation of a molecule in relation to its potential target, and it is also a crucial component of the rational drug design process [41]. In this study, among the seven isolated compounds, compounds **3** and **4** were chosen and docked into the binding site of the urease protein due to their significant anti-*H. pylori* and anti-urease properties. The molecular docking study aimed to identify the binding amino acid residues of these compounds to urease derived from *H. pylori* and to determine their respective binding energies. The results showed that the ligand molecules effectively bind to the target site. Specifically, compound **3** interacted with the Lys β445 and Gln β471 residues through hydrogen interaction, and with the Val α36, Val α33, Pro β472, and Leu α13 residues through hydrophobic interaction. Compound 4 interacted with the Gln β471, Gln β459, Tyr β32, Phe β441, Glu α80, Gly α82, Thr β469, and His α79 residues through hydrogen interaction, and with the Val β473 residue through hydrophobic interaction (Figure 4). Compound **4** showed lower binding energy compared to compound **3**. These findings from the molecular docking analysis are in good agreement with the in vitro urease inhibitory activity of compounds **3** and **4**. This suggests that both compounds have the potential to inhibit urease, which may contribute to their anti-*H. pylori* efficacy. Additionally, there can be other inhibitory mechanisms, considering that compound **3** exhibited more potent anti-*H. pylori* activity than compound **4**.

To assess the accuracy of docking program the co-crystallized ligand was removed from the active site and re-docked within the inhibitor binding cavity of *H. pylori* urease. In this study, the RMSD value was found to be 0.1 Å, showing that our docking method is valid for the studied inhibitors. The grid box parameters were centered on the native ligand’s coordinates.

These results provide a molecular rationale for the urease inhibitory activities observed in compounds **3** and **4** and highlight their potential as lead scaffolds for the development of anti-*H. pylori* agents.

## 3. Materials and Methods

### 3.1. General Experimental Procedures

The ^1^H and ^13^C-NMR spectra were recorded on a DRX-300 spectrometer, Avance Neo 400 nanobay, and Avance Neo 600 (Bruker, Billerica, MA, USA), and chemical shifts were measured as *δ* (ppm) values. All NMR solvents were purchased from Eurisotop (Tewksbury, MA, USA). QTOF mass spectrometry was performed using a Xevo G2-XS TOF (Waters, Framingham, MA, USA) and an X500R (AB SCIEX, Framingham, MA, USA) mass spectrometer. Experimental ECD spectra in EtOH were acquired in a quartz cuvette of 1 mm optical path length on a JASCO J-1500 spectropolarimeter (JASCO, Tokyo, Japan). Column chromatography (CC) was performed with Sephadex LH-20 (Cytiva, Uppsala, Sweden). Medium-pressure liquid chromatography (MPLC) was performed using a SNAP Cartridge KP-SIL 340 g (Biotage, Uppsala, Sweden). HPLC analysis was conducted with an Agilent 1260 HPLC system (Hewlett Packard, Waldbronn, Germany) equipped with a Gemini C18 column (110 Å, 4.6 × 250 mm, 5 µm) (Phenomenex, Torrance, CA, USA). A CO_2_ incubator (Sanyo, Sakata, Japan) was employed for the cultivation of *H. pylori*. All standard materials were purchased from Sigma-Aldrich (St. Louis, MO, USA), while other solvents were obtained from Fisher Scientific (Hampton, NH, USA).

### 3.2. Plant Materials

The *Sambucus williamsii* branch was collected from Hadong, Gyeongsangnam-do, Republic of Korea. The plant material was authenticated by Dr. Mi-Jeong Ahn (College of Pharmacy, Gyeongsang National University, Republic of Korea), and a specimen (No. PGSC-610) was deposited in the herbarium of College of Pharmacy, Gyeongsang National University.

### 3.3. Extraction and Isolation

The air-dried branch of *S. williamsii* (1.5 kg) was finely ground and extracted with methanol at room temperature. The extract was filtered and concentrated through a rotary evaporator. The resulting crude extract was then suspended in water and partitioned with *n*-hexane, methylene chloride (CH_2_Cl_2_), ethyl acetate (EtOAc) and *n*-butanol (BuOH) successively to yield a hexane fr. (9.2 g), CH_2_Cl_2_ fr. (6.0 g), EtOAc fr. (1.8 g), BuOH fr. (6.8 g) and aqueous fr. (11.8 g).

The CH_2_Cl_2_ fraction was subjected to MPLC using different solvents of increasing polarity (hexane, ethyl acetate, and methanol, 100:0:0 → 0:100:0 → 0:0:100), resulting in the collection of nine fractions (Fr. 1–Fr. 9). Fr. 7 was subjected to MPLC using a mixture of water and acetonitrile as an eluting solvent to provide seven subfractions (Fr. 7.1–Fr. 7.7). Fr. 7.1 was subjected to MPLC with a gradient of hexane, ethyl acetate, and methanol (100:0:0 → 0:100:0 → 0:0:100) as the eluting solvents, yielding 11 subfractions (Fr. 7.1.1–Fr. 7.1.11). Compounds **1** (1.7 mg, *t*_R_ 6.2 min) and **2** (0.9 mg, *t*_R_ 11.9 min) were isolated from Fr. 7.1.2 by prep-HPLC. The prep-HPLC separation was performed with an Agilent 1260 HPLC system with a Gemini C18 column. The chromatogram was performed by a gradient elution of water (A) and acetonitrile (B) mixture under the following conditions: 0 min, 15%; 0 to 5 min, 15% B; 5 to 40 min, 50% B; and 40 to 50 min, 95% B. The flow rate was 1 mL/min, and the temperature was maintained at 30 °C. Compound **3** (2.6 mg) was isolated from fr. 7.1.4 by performing Sephadex LH-20 column chromatography (CC) using methanol as the eluting solvent. Fr. 7.5 was subjected to Sephadex LH-20 CC using methanol as the eluting solvent, resulting in the isolation of compound **4** (1.3 mg). Fr. 8 was subjected to MPLC using a mixture of hexane, ethyl acetate, and methanol (100:0:0 → 0:100:0 → 0:0:100) as the eluting solvents. This process resulted in the collection of 10 subfractions (Fr. 8.1–Fr. 8.10). Fr. 8.5 was then subjected to Sephadex LH-20 CC with methanol as the eluting solvent, yielding six subfractions (Fr. 8.5.1–Fr. 8.5.6). A subfraction, Fr. 8.5.3, was further separated by prep-HPLC using the same HPLC system equipped with a Gemini C18 column. The gradient elution was accomplished as follows: 0 min, 10% B; 0 to 5 min, 10% B; 5 to 40 min, 20% B; 40 to 80 min, 50% B; 80 to 90 min, 95% B. This procedure resulted in the isolation of compounds **5** (1.9 mg, *t*_R_ 31.1 min), **6** (1.6 mg, *t*_R_ 33.2 min), and **7** (8.1 mg, *t*_R_ 50.3 min) (Appendix A).

(7*R*,8*R*)-Sambucanol (5) White amorphous powder; ${\left[\alpha \right]}_{D}^{20}$ −51 (*c* 0.16, EtOH); UV (MeOH) λ_max_ 205 (4.55), 225 (4.21), 250 (3.84), 270 (3.95), 290 (3.80), 335 (3.55, sh) nm; ECD (EtOH) λ_max_ (Δε) 251 (−0.06), 279 (−0.07) nm; ^1^H (CD_3_OD, 300 MHz) and ^13^C (CD_3_OD, 100 MHz) NMR data, see Table 2; ESI-QTOF-MS (*m*/*z*): 399.1379 [M + Na]^+^ (calcd. C_20_H_24_O_7_Na, 399.1420).

(7*S*,8*S*)-Guaiacylglycerol *β*-coniferyl ether (6) White amorphous powder; ${\left[\alpha \right]}_{D}^{20}$ −12 (*c* 0.13, MeOH); UV (MeOH) λ_max_ 205 (4.60), 225 (4.28), 250 (3.91), 270 (4.09), 290 (3.84), 335 (2.98, sh) nm; ECD (MeOH) λ_max_ (Δε) 217 (+0.06); ^1^H (CD_3_OD, 300 MHz) and ^13^C (CD_3_OD, 100 MHz) NMR data, see Table 2; ESI-QTOF-MS (*m*/*z*): 399.1379 [M + Na]^+^ (calcd. C_20_H_24_O_7_Na, 399.1420).

MS and NMR spectra for the isolated compounds **1**–**7**, see Appendix A.

### 3.4. ECD Calculation

Initial conformational searches were performed at the MMFF94 force field using the MacroModel (version 2021-4, Schrödinger LLC, New York, NY, USA) program with a mixed torsional/low-mode sampling method, in which a gas phase with a 50 kJ mol^−1^ energy window and 10,000 maximum iterations were employed. The Polak–Ribiere conjugate gradient protocol was established with 10,000 maximum iterations and a 0.001 kJ (mol Å)^−1^ convergence threshold on the root-mean-square gradient to minimize conformers. The conformers proposed in this study (found within 20 kJ mol^−1^ in the MMFF force field) were selected for geometry optimization using TmoleX 4.3.2 with the density functional theory settings of B3-LYP/6-31+G(d,p).

ECD calculations for the (7*R*,8*R*)-5 and (7*S*,8*S*)-5 conformers (13 conformers each) were performed at an identical theory level and basis sets. The (7*R*,8*R*)-6 and (7*S*,8*S*)-6 conformers were calculated, with 14 of each. The calculated ECD spectra were simulated by superimposing each transition, where σ is the bandwidth at a height of 1/e. Δ*E*_i_ and *R*_i_ are the excitation energy and rotatory strength for transition *i*, respectively. In this study, the value of σ was 0.2 eV. The excitation energies and rotational strengths of the ECD spectra were calculated based on the Boltzmann populations of conformers, and ECD visualization was performed using SigmaPlot 14.0.$$∆\in \left(E\right)=\frac{1}{2.297\;\times \;10^{-39}}\frac{1}{\sqrt{2\pi \sigma }}\sum_{A}^{i}∆E_{i}R_{i}e^{\left[-{\left(E-∆E_{i}\right)}^{2}/{(2\sigma )}^{2}\right]}$$

### 3.5. Helicobacter pylori Culture

The *H. pylori* strains 51 and 26,695 utilized in this study were acquired from the *Helicobacter pylori* Korean Type Culture Collection (HKTCC), Department of Microbiology, College of Medicine, Gyeongsang National University, Republic of Korea. Strain 51 was initially isolated from the stomach of a Korean patient in 1987 and 2000. Strain 26,695, known as KE26695, was identical to the strain isolated in the United Kingdom from the stomach of a patient with gastritis. The strains were cultured in Brucella Broth liquid medium (BD, Franklin Lakes, NJ, USA) supplemented with 10% horse serum (Gibco, Grand Island, NY, USA) and subcultured every 24 h at 37 °C under 10% CO_2_.

### 3.6. Anti-Helicobacter pylori Assay

The broth dilution method was used for determination of the minimal inhibitory concentration (MIC) values [42]. A bacterial colony suspension equivalent to 2–3 × 10^8^ cfu mL^−1^ and serial two-fold dilutions of each isolate and reference compound were prepared. The culture medium containing twenty microliters of bacterial inoculum and twenty microliters of sample solution was incubated at 37 °C for 24 h. After incubation, MIC, MIC_50_, and MIC_90_ were defined as the lowest concentration of compounds at which bacterial growth was inhibited, and inhibited by 50% and 90%, respectively. Analysis of growth was achieved by reading the optical density at 600 nm. All of the values were obtained from three independent experiments. The MIC_50_ and MIC_90_ values were calculated using GraphPad Version 5.01 (GraphPad Software, Inc., San Diego, CA, USA). DMSO was used as the negative control, while quercetin and metronidazole (Sigma-Aldrich, St. Louis, MO, USA) were used as the positive control. The following formula was used to calculate the inhibitory activity.Inhibition (%) = [(Absorbance of negative control − Absorbance of sample solution)/Absorbance of negative control] × 100

### 3.7. Measurement of Urease Inhibitory Activity

The anti-urease assay was conducted using phenol red reagent, referring to the previously reported paper [33]. The culture fluid of *H. pylori* strain 51 was centrifuged at 500× *g* for two minutes, and the pellet was resuspended in lysis buffer (Invitrogen, Middlesex County, MA, USA). After one minute of ultrasonic disruption, the suspension was incubated on ice for ten minutes. This process was carried out three times. After centrifuging the mixture for one minute at 100× *g*, 20 μL of the supernatant was combined with 50 μL of 10 μM urea and 20 μL of sample solution and incubated for ten minutes at 37 °C with 10% CO_2_. The final concentration of sample was 1 mM. The incubation was carried out for 30 min after the concentration was changed to 0.1 M urea and 1 mM phenol red. Absorbance was measured at 570 nm. Acetohydroxamic acid (Sigma-Aldrich, St. Louis, MO, USA) was used as a positive control [43]. The same formula for MIC values was used to calculate the urease inhibitory activity.

### 3.8. Molecular Docking

In the study, the AutoDock Vina tool (version 1.5.7) was used to investigate the molecular interaction between target protein and the selected ligand. The Crystallographic structure of *H. pylori* urease (PDB ID: 1E9Y) was taken from Protein Data Bank. Before docking analysis, BIOVIA Discovery Studio 2021 program used for the optimization of structure of the enzyme by removing excess ligands and water molecules. The selected compounds were drawn in ChemDraw 12.0 (Cambridgesoft, Cambridge, MA, USA) and then all compounds were optimized for energy using the Spartan 14 (Version 1.1.4) program. Polar hydrogens were added to the protein using the AutoDock vina 1.5.7 tool, and Kollman charges were determined using Compute Gasteiger. BIOVIA Discovery was used to determine the active sites of proteins. Finally, the molecular interactions and types of bond between the selected compound and target protein were investigated using the Discovery Studio visualizer programs.

### 3.9. Statistical Analysis

All experiments were performed in triplicate, and the results are presented as the mean ± standard deviation (SD). One-way analysis of variance (one-way ANOVA) was conducted using the SPSS Statistics 24.0 software (IBM, Armonk, NY, USA). The significance level was evaluated at * *p* < 0.05.

## 4. Conclusions

In this study, three phenolic compounds and four lignans were isolated from the CH_2_Cl_2_ fraction of *S. williamsii* branch extract. Among the isolated compounds, (7*R*,8*R*)-sambucanol (**5**) was reported here for the first time in nature. The compounds (7*S*,8*S*)-Guaiacylglycerol *β*-coniferyl ether (**6**) and lawsonicin (**7**) were also newly identified within the *Sambucus* species and Viburnaceae family, respectively. We also documented the anti-*H. pylori* and anti-urease activities of (7*S*,8*R*)-guaiacylglycerol (**3**) and boehmenan (**4**) for the first time. These findings provide new insight into the phytochemical diversity of *S. williamsii* and support its potential as a natural therapeutic agent against *H. pylori*. Furthermore, the stereochemical configurations of these compounds have been thoroughly determined for the first time, which contributes to our knowledge of their structural diversity and advances the field of natural product chemistry. The identification of (7*S*,8*R*)-guaiacylglycerol and boehmenan as key active components of *S. williamsii* branch opens up new avenues for future research into natural product-based treatments for *H. pylori* infections, particularly in light of growing concerns about antibiotic resistance and the side effects associated with conventional therapies. Further investigations will aim to isolate additional anti-*H. pylori* constituents from *S. williamsii* and elucidate their mechanisms of action through in vitro and in vivo assays and molecular docking studies for the design of more potent natural products for eradicating *H. pylori*.

## Figures and Tables

**Figure 1 plants-14-02558-f001:**
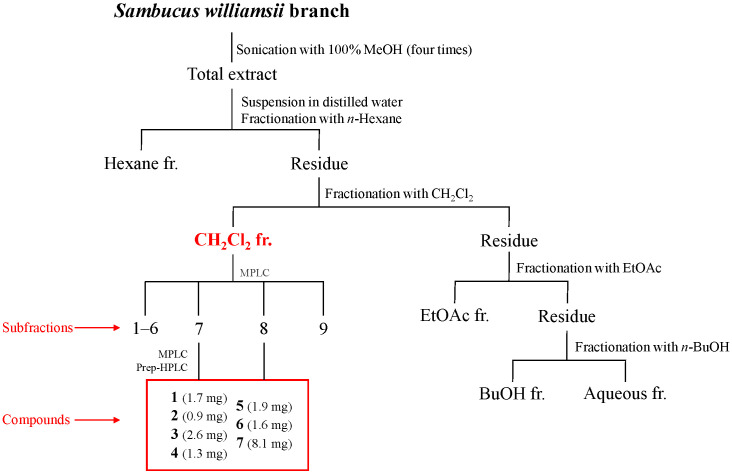
Isolation of compounds **1**–**7** from *Sambucus williamsii* branch.

**Figure 2 plants-14-02558-f002:**
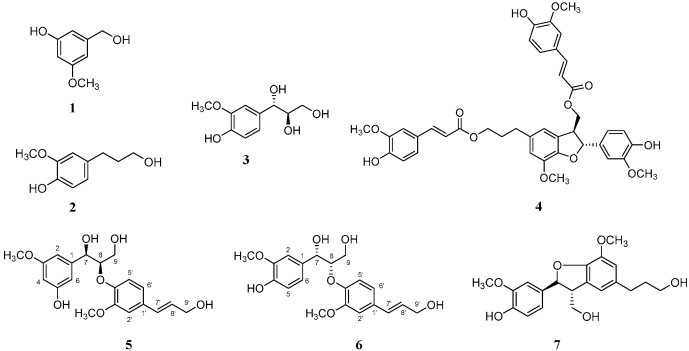
Chemical structures of compounds **1**–**7**.

**Figure 3 plants-14-02558-f003:**
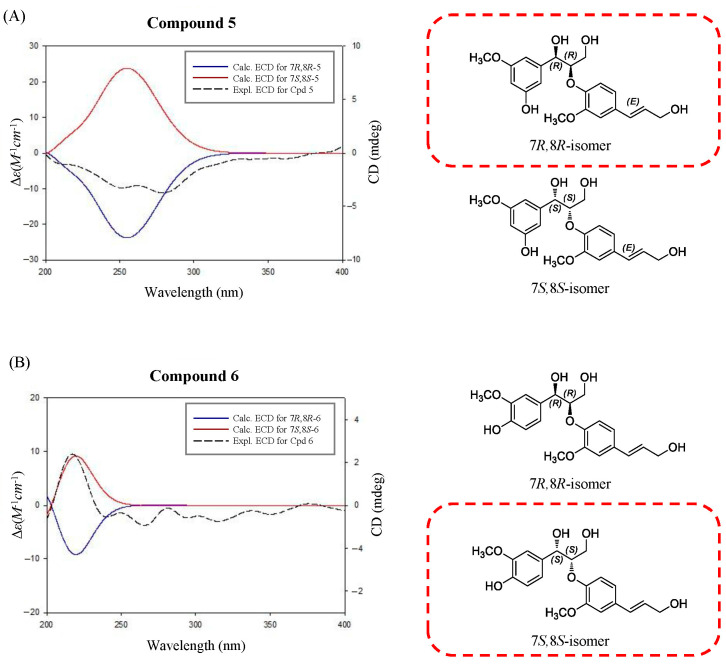
Comparison of the experimental ECD spectra of compounds **5** (**A**) and **6** (**B**) with the calculated ECD spectra of their two possible stereoisomers, (7*R*,8*R*)- and (7*S*,8*S*)-isomers.

**Figure 4 plants-14-02558-f004:**
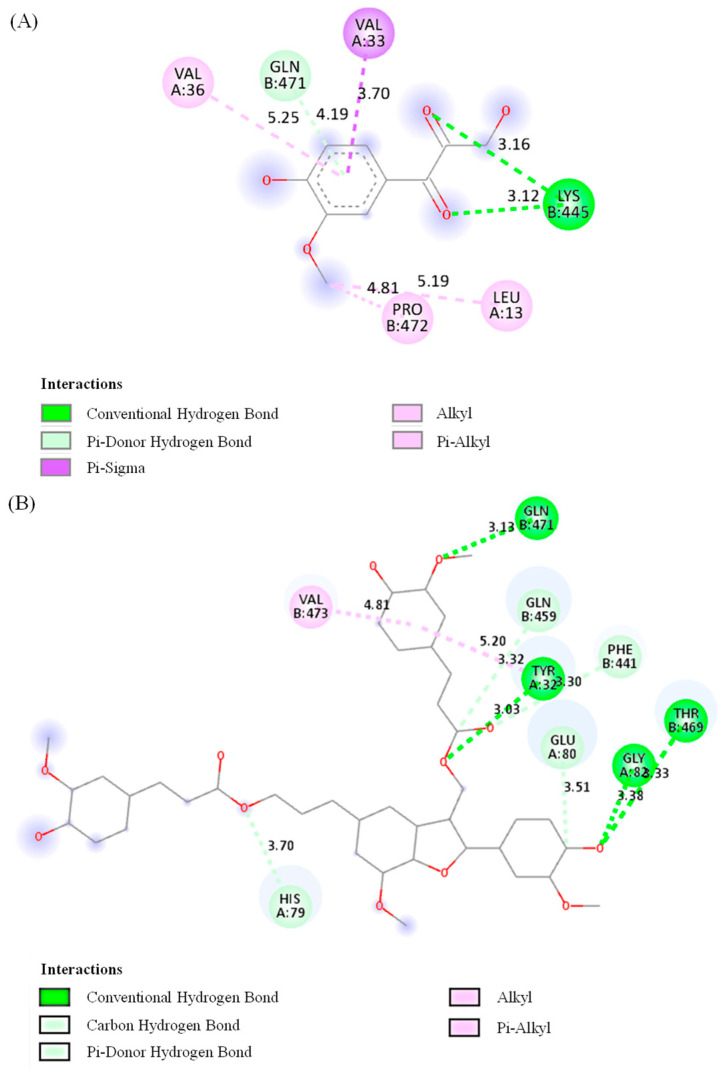
Two-dimensional interaction diagrams of two compounds **3** (**A**) and **4** (**B**) with *H. pylori* urease. Pink color depicts alkyl or pi–alkyl interaction. In alkyl interaction, Van der Waals forces stabilize the complex through a hydrophobic contact between the ligand and alkyl side chains (such as -CH_3_, -CH_2_– groups) of amino acid residues. The pi–alkyl interaction is a hydrophobic contact that improves ligand–receptor packing between alkyl groups and the π-electron system of an aromatic ring. Through close-range interactions, the π-system of an aromatic ring and a sigma (σ) bond interact to produce the pi–sigma interaction, shown in purple. A conventional hydrogen bond is expressed in green. It is an essential component of ligand binding specificity and affinity, and a traditional hydrogen bond involves a shared hydrogen atom between a donor (such as -OH, -NH_2_) and an acceptor (such as O and N). The carbon hydrogen bond depicted in light green describes the interaction between hydrogen atoms and carbon atoms (carbanions from the neighboring electronegative atoms) present in the ligand. The pi–donor hydrogen bond in light green represents a hydrogen bond in which an electronegative acceptor atom in the protein interacts with an aromatic π-system, which functions as a hydrogen bond donor.

**Table 1 plants-14-02558-t001:** Anti-*Helicobacter pylori* activity of *S. williamsii* fractions.

Strains	Total Fr.	Hexane Fr.	CH_2_Cl_2_ Fr.	EtOAc Fr.	BuOH Fr.	Aqueous Fr.	Quercetin
51	24.3 ± 3.7 ^b^	53.0 ± 3.7 ^a^	69.8 ± 3.7 ^a^	36.4 ± 0.9 ^a^	26.7 ± 3.7 ^b^	23.9 ± 2.1 ^b^	43.2 ± 9.6 ^a^
26,695	18.5 ± 15.1 ^b^	20.0 ± 14.3 ^b^	94.7 ± 1.3 ^a^	92.5 ± 1.4 ^a^	12.0 ± 1.6 ^b^	5.3 ± 7.5 ^c^	24.8 ± 7.1 ^b^

The inhibitory activity (%) was expressed as the mean ± SD (*n* = 3). Different superscript letters in the same row indicate statistically significant differences among groups according to Duncan’s multiple-range test (*p* < 0.05).

**Table 2 plants-14-02558-t002:** ^1^H (300 MHz) and ^13^C-NMR (100 MHz) data of compounds **5** and **6** in CD_3_OD (*δ* ppm).

Position	5		6	
	*δ*_H_ (*J* in Hz)	*δ* _C_	*δ*_H_ (*J* in Hz)	*δ* _C_
1		130.1		130.0
2	6.88 d (1.4)	119.6	7.08 d (1.9)	110.3
3		147.5		150.4
4	7.01 d (1.4)	119.3		147.6
5		147.6	6.76 d (8.1)	119.4
6	7.01 d (1.4)	110.4	6.86 dd (8.1, 1.9)	117.5
7	4.84 d (5.7)	72.7	4.89 d (5.7)	72.7
8	4.37 td (5.7, 3.8)	84.8	4.31 td (5.7, 3.9)	85.8
9	3.85 d (8.1, 5.7, 9a)	60.8	3.75 dd (11.9, 3.9, 9a)	60.5
	3.80 d (8.1, 3.8, 9b)		3.48 dd (11.9, 5.7, 9b)	
1′		131.6		131.7
2′	6.88 d (1.4)	110.0	7.03 d (1.9)	109.9
3′		150.5		150.4
4′		146.2		147.9
5′	6.73 d (8.1)	114.4	7.02 d (8.1)	114.5
6′	6.83 dd (8.1, 1.4)	117.5	6.93 dd (8.1, 1.9)	119.4
7′	6.53 dt (15.9, 1.4)	132.3	6.56 d (15.9)	132.1
8′	6.25 td (15.9, 5.7)	127.1	6.28 dt (15.9, 8.1)	127.2
9′	4.21 dd (5.7, 1.4)	62.4	4.22 dd (5.7, 1.4)	62.4
3-OCH_3_	3.82 s	55.9	3.90 s	55.13
3′-OCH_3_	3.81 s	55.1	3.83 s	54.90

**Table 3 plants-14-02558-t003:** Anti-*Helicobacter pylori* activity of compounds **1**–**7**.

Strains	1	2	3	4	5	6	7	Quercetin	Metronidazole
51	69.0 ± 1.1 ^c^	12.3 ± 5.8 ^e^	94.5 ± 0.3 ^a^	81.0 ± 1.0 ^b^	7.1 ± 2.4 ^e^	13.1 ± 3.9 ^e^	21.6 ± 0.9 ^d^	35.7 ± 3.5 ^d^	95.5 ± 0.4 ^a^
26,695	72.0 ± 1.2 ^c^	4.0 ± 0.9 ^e^	97.3 ± 0.1 ^a^	85.0 ± 3.2 ^b^	ND	1.7 ± 1.5 ^d^	2.6 ± 1.5 ^e^	20.2 ± 2.7 ^d^	94.6 ± 0.2 ^a^

The inhibitory activity (%) was expressed as the mean ± SD (*n* = 3). Different superscript letters in the same row indicate statistically significant differences among groups according to Duncan’s multiple range test (*p* < 0.05).

**Table 4 plants-14-02558-t004:** Anti-*H. pylori* activity of compounds **3** and **4**.

Strains	MIC (μM) ^a^	3	4	Quercetin ^b^	Metronidazole ^b^
51	MIC	3.13	3.13	50	3.13
	MIC_50_	28.5	66.0	>100	13.7
	MIC_90_	97.1	>100	>100	46.5
26,695	MIC	6.25	6.25	50	3.13
	MIC_50_	56.8	62.0	>100	18.9
	MIC_90_	86.5	>100	>100	40.7

^a^ MIC, MIC_50_, and MIC_90_ values of compounds toward *H. pylori* were determined as the lowest concentration (μM) that resulted in in vitro bacterial growth inhibition: 50% and 90% inhibition, respectively. ^b^ Quercetin and metronidazole served as positive controls.

**Table 5 plants-14-02558-t005:** Binding energies, interacting residues, and inhibitory activity of ligand compounds **3** and **4** against *H. pylori* urease protein.

Compounds	Binding Energy ^a^	Hydrogen Interaction ^b^	Hydrophobic Interaction ^c^	Inhibition (%) ^d^
(7S,8R)-Guaiacylglycerol (**3**)	−6.0	Lys β445, Gln β471	Val α36, Val α33, Pro β472, Leu α13	31.8 ± 2.8 ^f^
Boehmenan (**4**)	−8.6	Gln β471, Gln β459, Tyr β32, Phe β441, Glu α80, Gly α82, Thr β469, His α79	Val β473, Tyr α32	38.9 ± 1.2 ^e^

^a, b, c^ These parameters were obtained from molecular docking simulation. ^d^ The inhibitory activity was evaluated via an in vitro assay with *H. pylori* urease. ^e,f^ Different letters in the same column indicate a significant difference according to Duncan’s multiple-range test (*p* < 0.05).

## Data Availability

The original contributions presented in this study are included in the article/Appendix A. Further inquiries can be directed to the corresponding author.

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
