# Peer review of "Anti-Helicobacter pylori Compounds of Sambucus williamsii Hance Branch"

_plants, 2025, doi:10.3390/plants14162558_

Round 1
Reviewer 1 Report
Comments and Suggestions for Authors
This manuscript presents the isolation of metabolites from Sambucus williamsii and the in silico and in vitro evaluation against H. pylori. As a result, seven metabolites were isolated, two of them undescribed in literature. Compounds 3 and 4 displayed moderate urease inhibitory activity, suggesting their potential to inhibit the enzyme and contribute to their overall anti-H. pylori efficacy
.
Thereby, I recommend to publish this paper after major revisions as followings:
- Table 1 contains errors in the assignment of 1H and 13C NMR data. For instance, the 1H and 13C NMR data for position 1 in compounds 5 and 6 are interchanged. Additionally, 1H NMR data for positions 9a, 9b, 9’a, and 9’b, and 13C NMR data for positions 9 and 9’ in compound 5 are missing. The same issues are observed for compound 6.
Author Response
Response to the Reviewer 1
Thank you very much for your consideration. We have considered your comments very closely in revising this manuscript as follows.
Comments:
This manuscript presents the isolation of metabolites from Sambucus williamsii and the in silico and in vitro evaluation against H. pylori. As a result, seven metabolites were isolated, two of them undescribed in literature. Compounds 3 and 4 displayed moderate urease inhibitory activity, suggesting their potential to inhibit the enzyme and contribute to their overall anti-H. pylori efficacy. Thereby, I recommend to publish this paper after major revisions as followings.
1) Table 1 contains errors in the assignment of 1H and 13C NMR data. For instance, the 1H and 13C NMR data for position 1 in compounds 5 and 6 are interchanged. Additionally, 1H NMR data for positions 9a, 9b, 9ʹa, and 9ʹb, and 13C NMR data for positions 9 and 9ʹ in compound 5 are missing. The same issues are observed for compound 6.
Response: The 1H and 13C NMR data for position 1 in compounds 5 and 6 has now been corrected. The 1H and 13C NMR assignments for positions 9a, 9b, 9ʹa, and 9′b were re-examined for both compounds, and the other errors were corrected (in Table 2).
Thank you for your comment improving our manuscript’s accuracy.

Reviewer 2 Report
Comments and Suggestions for Authors
This study comprehensively investigates the anti-Helicobacter pylori effects of compounds obtained from Sambucus williamsii branches and leads to the identification of new natural compounds effective against this organism. The isolation, structural characterization (NMR, MS, ECD), and biological activities (MIC, urease inhibition, and molecular docking) of seven different compounds are presented in detail. In particular, the reporting of (7S,8R)-guaiacylglycerol and boehmenan as both anti-H. pylori and urease inhibitors enhances the originality of the study. Overall, the article is methodologically sound, the writing is clear, and the data is supported by graphs and tables.
However, some sections need to be more clearly stated, and several corrections are necessary for the completeness of the data.
1- Statistical significance is shown for the data presented in Tables 1, 3, and 4 (e.g., Duncan’s test). However, the groups between which the significant differences are are not clearly stated in the table notes. This detail will facilitate the reader's interpretation of the data.
2- The urease inhibition test protocol is only summarized. The positive control used is not specified. This prevents comparative analysis. Furthermore, percentage (%) is used instead of inhibitor concentration (μM), which limits comparative interpretations.
3- For example, the term "subfractions" is used repeatedly, but it is unclear which compounds were isolated from how many of them. Matching the codes of these subfractions and the isolated compounds should be supported with a diagram or supplementary table.
Some statements in the article are repetitive or unclear. For example, the phrase "Compound 4 also showed inhibitory effects" is already stated in the previous sentence.
4- Although binding energies are presented in the molecular docking results, RMSD values are not. This makes it difficult to assess the reliability of the docking scores.
Furthermore, details such as the "grid box" parameters used during docking and the validation strategy (e.g., re-docking) are not provided.
5- In particular, no comparisons were made with previous studies on the anti-H. pylori effects of lignans. The contextual significance of the results can be emphasized by comparing them with the effects of similar compounds in the literature.
6- The descriptions of the ECD and docking results, particularly in Figures 3 and 4, are weak and should be supported with descriptive titles and figure captions.
7- A more robust discussion of how the study will contribute to future clinical applications or natural product-based drug development is anticipated.
Author Response
Response to the Reviewer 2
Thank you very much for your consideration. We have considered your comments very closely in revising this manuscript as follows.
Comments:
This study comprehensively investigates the anti-Helicobacter pylori effects of compounds obtained from Sambucus williamsii branches and leads to the identification of new natural compounds effective against this organism. The isolation, structural characterization (NMR, MS, ECD), and biological activities (MIC, urease inhibition, and molecular docking) of seven different compounds are presented in detail. In particular, the reporting of (7S,8R)-guaiacylglycerol and boehmenan as both anti-H. pylori and urease inhibitors enhances the originality of the study. Overall, the article is methodologically sound, the writing is clear, and the data is supported by graphs and tables. However, some sections need to be more clearly stated, and several corrections are necessary for the completeness of the data.
1) Statistical significance is shown for the data presented in Tables 1, 3, and 4 (e.g., Duncan’s test). However, the groups between which the significant differences are not clearly stated in the table notes. This detail will facilitate the reader’s interpretation of the data.
Response: We have revised the footnotes of Tables 1 and 3 to clearly indicate the statistical comparison among groups. The revised footnote is “Different superscript letters in the same row indicate statistically significant differences among groups according to Duncan’s multiple range test (p < 0.05).”
2) The urease inhibition test protocol is only summarized. The positive control used is not specified. This prevents comparative analysis. Furthermore, percentage (%) is used instead of inhibitor concentration (μM), which limits comparative interpretations.
Response: The inhibitory activity of a positive control, acetohydroxamic acid was added with the related content in the Results and Discussion Section (lines 239-240), and the more detail experimental protocol was added in the Materials and Methods Section (lines 447-448, lines 453 and 454, and lines 456 and 457). The added descriptions were as follows: The anti-urease assay was conducted using phenol red reagent referring to the previously reported paper [33]. … with 50 μL of 10 μM urea and 20 μL of sample solution, and … The final concentration of sample was 1 mM. … Acetohydroxamic acid (Sigma-Aldrich, St. Louis, MO, USA) was used as a positive control.
3) For example, the term “subfractions” is used repeatedly, but it is unclear which compounds were isolated from how many of them. Matching the codes of these subfractions and the isolated compounds should be supported with a diagram or supplementary table. Some statements in the article are repetitive or unclear. For example, the phrase “Compound 4 also showed inhibitory effects” is already stated in the previous sentence.
Response: In order to make the isolation scheme clearer than Figure 1, we have added a supplementary diagram (Figure S1) that summarizes the connection between the subfractions mentioned in the Materials and Methods Section and the corresponding isolated compounds.
4) Although binding energies are presented in the molecular docking results, RMSD values are not. This makes it difficult to assess the reliability of the docking scores. Furthermore, details such as the “grid box” parameters used during docking and the validation strategy (e.g., re-docking) are not provided.
Response: To clarify the reliability of the docking scores, the RMSD value, grid box parameters, and re-docking strategy were mentioned in the subsection of Molecular Docking Simulation as follows: To assess the accuracy of docking program, the co-crystallized ligand was removed from the active site and re-docked within the inhibitor binding cavity of pylori urease. In this study, RMSD value was found as 0.1 Å, showing that our docking method is valid for the studied inhibitors. The grid box parameters were centered on the native ligand’s coordinates (lines 315-319).
5) In particular, no comparisons were made with previous studies on the anti-H. pylori effects of lignans. The contextual significance of the results can be emphasized by comparing them with the effects of similar compounds in the literature.
Response: We could not find any significant anti-H. pylori lignan compounds from previous studies. This fact can come from the fact that not many studies have been conducted to find anti-H. pylori compounds from natural resources.
6) The descriptions of the ECD and docking results, particularly in Figures 3 and 4, are weak and should be supported with descriptive titles and figure captions.
- Response: The legends of Figures 3 and 4 have been changed into as follows:
- Figure 3. Experimental and calculated ECD spectra of compounds 5 (A) and 6 (B). à Figure 3. Comparison of the experimental ECD spectra of compounds 5 (A) and 6 (B) with the calculated ECD spectra of their two possible stereoisomers, (7R,8R)- and (7S,8S)-isomers.
- Figure 4. 2D interaction diagrams of two compounds 3 (A) and 4 (B) with H. pylori urease. à Figure 4. 2D interaction diagrams of two compounds 3 (A) and 4 (B) with H. pylori urease. Pink in color depicts alkyl or pi-alkyl interaction. In alkyl interaction, Van der Waals forces stabilize the complex through a hydrophobic contact between the ligand and alkyl side chains (such as -CH₃, -CH₂– groups) of amino acid residues. The pi-alkyl interaction is a hydrophobic contact that im-proves ligand-receptor packing between alkyl groups and the π-electron system of an aromatic ring. Through close-range interactions, the π-system of an aromatic ring and a sigma (σ) bond interact to produce the pi-sigma interaction, in purple in color. Conventional hydrogen bond was expressed in green color. It is an essential component of ligand binding specificity and affinity, and a traditional hydrogen bond involves a shared hydrogen atom between a donor (such as -OH, -NH₂) and an acceptor (such as O and N). Carbon hydrogen bond in light green describes the interaction between hydrogen atoms and carbon atoms (carbanions from the neighboring electronegative atoms) present in ligand. Pi-donor hydrogen bond in light green represents a hydrogen bond in which an electronegative acceptor atom in the protein interacts with an aromatic π-system, which functions as a hydrogen bond donor.
7) A more robust discussion of how the study will contribute to future clinical applications or natural product-based drug development is anticipated.
- Response: A sentence on future studies was added as the last sentence of the Conclusion Section as follows: Further investigations will aim to isolate additional anti- pylori constituents from S. williamsii and elucidate their mechanisms of action through in vitro and in vivo assays and molecular docking studies for the design of more potent natural products for eradicating H. pylori (lines 501-504).
Thanks again for your comments improving our manuscript.

Reviewer 3 Report
Comments and Suggestions for Authors
1- Abstract line 27:.........the structure of was established by 1D and 2D NMR.... as mentioned by author..... but I did not find HMBC, HMQC, COSY 2D NMR spectra, which are necessary for the structure elucidation of compound, and essential for new compound structural proofs.
2- Author need to explain all 2D NMR correlations by which new compound structure has been elucidated.
3- Kindly submit the HPLC chromatograms as supplementary file.
Author Response
Response to the Reviewer 3
Thank you very much for your consideration. We have considered your comments very closely in revising this manuscript as follows.
Comments:
1- Abstract line 27:.........the structure of was established by 1D and 2D NMR.... as mentioned by author..... but I did not find HMBC, HMQC, COSY 2D NMR spectra, which are necessary for the structure elucidation of compound, and essential for new compound structural proofs.
2- Author need to explain all 2D NMR correlations by which new compound structure has been elucidated.
- Response: The description of “by 1D and 2D NMR” was changed into “1H- and 13C-NMR”. We tried to get the 2D NMR for a new compound (5) during our manuscript preparation, but failed to get clear correlation peaks due to its very low quantity and weight loss from ECD measurement and biological activity tests. At first, we did not realize the necessity of getting all 2D-NMR of compound 5 because the chemical structure of compound 5 is quite similar to the previously reported compound, and the NMR data are also similar, as mentioned in the context. In addition, stereoselectivity could be obtained from ECD experiments.
3- Kindly submit the HPLC chromatograms as supplementary file.
- Response: The HPLC chromatograms for the isolated compounds, including a new compound, 5, were added in the Supplementary information file as Figure S23 with the HPLC conditions.
Please consider our experimental situation. Thank you.

Reviewer 4 Report
Comments and Suggestions for Authors
Jeong and co-authors have investigated the phytochemical composition of Sambucus williamsii branches. Altogether 7 phenolic compounds were isolated from the dichlormethane extract, including one new compound. The structures were elucidated based on NMR, mass and circular dichroism data. In addition the compounds were tested for their antibacterial effect against Helicobacter pylori, with some additional experiments on the urease inhibition activity of two isolated compounds. In a supplementary part of the manuscript NMR spectra and further spectra are depicted.
The authors address an urgent need for new drugs against H. pylori. The whole article is written in an excellent and fluid style and the elucidation of two chemical structures are presented in a precise way and in full detail. 40 references are cited.
HOWEVER, there are some issues which need to be addressed and corrected by the authors:
- Lines 130, 370 and 477: Please write the botanical names and the name of the bacterium in italics!
- Line 153: Please specify how the identification was performed (NMR?, mass data? Values from the literature and/or a database?)?
- Line 202 / Table 4: and also lines 345-347 Please specify what „MIC“ means! Do you mean MIC100! Why is the value for the inhibition of MIC50 (50 % inhbition) smaller than the value for MIC90 (90 % inhibition)? Do you have an explanation or should „MIC90“ be „MIC10“ (i.e. 10 % inhibition)? Could you please check this again!
- Lines 213 and 217: it sounds a bit contradictory to have a significant effect, which is only moderate. Please check this again!
- Line 225: it should be 2.4
- Line 260 please add „column“
- Line 310: Please add that the compounds 1-4 were structurally elucidated based on NMR?, mass data? Literature data and/or a database (which database?)!
- Line 371: optimization
- Line 385: „seven compounds of“ can be ommitted. Otherwise the sentence should be rephrased.
- Please add a sentence on future studies (isolation of more compounds?, in vivo-testing? etc.)
To sum up this is an excellent manuscript which combines a phytochemical and a pharmacological study.
Author Response
Response to the Reviewer 4
Thank you very much for your consideration. We have considered your comments very closely in revising this manuscript as follows.
Comments:
Jeong and co-authors have investigated the phytochemical composition of Sambucus williamsii branches. Altogether 7 phenolic compounds were isolated from the dichlormethane extract, including one new compound. The structures were elucidated based on NMR, mass and circular dichroism data. In addition the compounds were tested for their antibacterial effect against Helicobacter pylori, with some additional experiments on the urease inhibition activity of two isolated compounds. In a supplementary part of the manuscript NMR spectra and further spectra are depicted.
1) Lines 130, 370 and 477: Please write the botanical names and the name of the bacterium in italics!
- Response: The botanical names and the bacterial name have been corrected to be italics (lines 130, 462, and 583).
2) Line 153: Please specify how the identification was performed (NMR?, mass data? Values from the literature and/or a database?)?
- Response: The identification was accomplished by comparison of their spectroscopic data, such as MS and NMR, with the published literature, and the related description has been added into the sentence (lines 164 and 165). In addition, individual references have been provided for each compound to enhance clarity.
3) Line 202 / Table 4: and also lines 345-347 Please specify what “MIC” means! Do you mean MIC100! Why is the value for the inhibition of MIC50 (50 % inhbition) smaller than the value for MIC90 (90 % inhibition)? Do you have an explanation or should “MIC90” be “MIC10” (i.e. 10 % inhibition)? Could you please check this again!
- Response: Three MIC values of MIC, MIC50 and MIC90 values presented in Table 4 are based on conventional definitions. MIC means the lowest concentration at that a sample starts the bacterial growth. MIC50 and MIC90 refer to the minimal inhibitory concentrations required to inhibit 50 percent and 90 percent of bacterial growth, respectively. Therefore, it is expected that MIC90 values are bigger than MIC50 values, which is consistent with the trend observed in our data. The relevant reference was added to the Materials and Methods Section (line 432).
4) Lines 213 and 217: it sounds a bit contradictory to have a significant effect, which is only moderate. Please check this again!
- Response: The former term “significant” has been changed into “moderate” to enhance the consistency (line 237).
5) Line 225: it should be 2.4
- Response: Corrected as suggested (line 250). Thank you.
6) Line 260 please add “column”
- Response: Corrected as suggested (line 336).
7) Line 310: Please add that the compounds 1-4 were structurally elucidated based on NMR?, mass data? Literature data and/or a database (which database?)!
- Response: The identification was accomplished by comparison of their spectroscopic data, such as MS and NMR, with the published literature, and the sentence of “MS and NMR spectra for the isolated compounds 1−7, see Figures S2−S22.” was added (line 400).
8) Line 371: optimization
- Response: The typographical error has been corrected as “optimization” (line 463).
9) Line 385: “seven compounds of” can be ommitted. Otherwise the sentence should be rephrased.
- Response: Corrected as suggested (line 487). Thank you.
10) Please add a sentence on future studies (isolation of more compounds?, in vivo-testing? etc.)
- Response: A sentence on future studies was added as the last sentence of the Conclusion Section as follows: Further investigations will aim to isolate additional anti- pylori constituents from S. williamsii and elucidate their mechanisms of action through in vitro and in vivo assays and molecular docking studies for the design of more potent natural products for eradicating H. pylori (lines 501-504). Another sentence was added to the Results and Discussion Section (lines 210-213).
Thanks again for your considerable comments improving our manuscript.

Round 2
Reviewer 1 Report
Comments and Suggestions for Authors
I consider that manuscript can be published
Reviewer 3 Report
Comments and Suggestions for Authors
Accept